# Melanisation in Boreal Lichens Is Accompanied by Variable Changes in Non-Photochemical Quenching

**DOI:** 10.3390/plants11202726

**Published:** 2022-10-15

**Authors:** Nqobile Truelove Ndhlovu, Knut Asbjørn Solhaug, Farida Minibayeva, Richard Peter Beckett

**Affiliations:** 1School of Life Sciences, University of KwaZulu-Natal, Private Bag X01, Scottsville 3209, South Africa; 2Faculty of Environmental, Sciences and Natural Resource Management, Norwegian University of Life Sciences, P.O. Box 5003, 1432 Ås, Norway; 3Kazan Institute of Biochemistry and Biophysics, Federal Research Center “Kazan Scientific Center of RAS”, P.O. Box 30, 420111 Kazan, Russia; 4Open Lab ‘Biomarker’, Kazan (Volga Region) Federal University, Kremlevskaya Str. 18, 420008 Kazan, Russia

**Keywords:** lichens, non-photochemical quenching, melanins, photoprotection, photoinhibition

## Abstract

Lichens often grow in microhabitats where they absorb more light than they can use for fixing carbon, and this excess energy can cause the formation of harmful reactive oxygen species (ROS). Lichen mycobionts can reduce ROS formation by synthesizing light-screening pigments such as melanins in the upper cortex, while the photobionts can dissipate excess energy radiationlessly using non-photochemical quenching (NPQ). An inherent problem with using fluorimetry techniques to compare NPQ in pale and melanised thalli is that NPQ is normally measured through a variously pigmented upper cortex. Here we used a dissection technique to remove the lower cortices and medullas of *Lobaria pulmonaria* and *Crocodia aurata* and then measure NPQ from the underside of the thallus. Results confirmed that NPQ can be satisfactorily assessed with a standard fluorimeter by taking measurement from above using intact thalli. However, photobionts from the bottom of the photobiont layer tend to have slightly lower rates of PSII activity and lower NPQ than those at the top, i.e., display mild “shade” characteristics. Analysis of pale and melanised thalli of other species indicates that NPQ in melanised thalli can be higher, similar or lower than pale thalli, probably depending on the light history of the microhabitat and presence of other tolerance mechanisms.

## 1. Introduction

When growing in exposed microhabitats photosynthetic organisms tend to absorb more light than they can use for fixing carbon, which can result in reductions in photosynthetic activity, often termed “photoinhibition” that will eventually reduce growth. Most authors believe that photoinhibition occurs when this excess energy results in the production of reactive oxygen species (ROS) [1,2]. ROS can cause lipid peroxidation or damage the D1 and D2 proteins in the reaction centre of PSII [3]. Like other photosynthetic organisms, it seems likely that lichens possess a range of mechanisms to cope with both long- and short-term changes in light availability (for review see [4]), although there are undoubtedly overlaps between the two sets of mechanisms. Many lichens grow on tree trunks or on rocks under a tree canopy, and in such microhabitats experience rapidly changing light levels. This is a result of gaps in the canopy creating brief periods of high light, known as “sunflecks”. Tolerance to these short-term changes in light availability (over a range from minutes to hours) can be improved by increasing the dissipation of excess energy harmlessly as heat using non-photochemical quenching (NPQ). Quenching has two main components; the first is “energy dependent quenching”, which is quenching that is relaxed during the first 200 s of darkness with a relaxation halftime of c. 30 s. This parameter is influenced by low lumen pH and the xanthophyll cycle [5]. The second component is photoinhibitory quenching and represents the fraction of quenching that takes place from about 8–10 min or longer to relax. Despite its name, it is now realized that photoinhibition is only one of the many processes responsible for slow relaxing quenching, and probably includes, for example, zeaxanthin-dependent slow-relaxing quenching [6]. It is worth noting that in addition, higher NPQ also protects plants against long-term changes in light levels. Higher “sun” plants typically have three to four-fold larger pools of xanthophyll cycle pigments than “shade” plants, and as a result have higher NPQ [7]. However, when *Arabidopsis* is exposed to an identical total dosage of light, supplied under either constant or fluctuating conditions, it is the plants receiving fluctuating light that display highest NPQ [8]. It seems likely that the high NPQ protects *Arabidopsis* against the brief periods of high light that occur during sunflecks. For lichens, species that grow in microhabitats where much of the light is derived from sunflecks usually display considerably higher NPQ than collections from more open microhabitats [9,10]. Although little studied in lichens, additional short-term mechanisms may include an increased ability to scavenge ROS formed during photoinhibition, an increased capacity to repair ROS-induced damage, and an increase in cyclic electron flow [11].

Lichens growing in “exposed” habitats may experience more sustained light stress, and the stress may vary only seasonally. Tolerance mechanisms to long-term (over a range from weeks to months) and possibly more pronounced light stress probably include the photobiont regulating light harvesting complex proteins or changing the ratio of PSII to PSI [12]. However, a particular tolerance mechanism for long-term light stress that has been well studied in lichens is the synthesis of secondary metabolites in the upper cortex. Some of these compounds, for example melanins, are intensely pigmented, and directly absorb PAR and UV light. As discussed in our recent review [12], the trigger for melanisation is UV, particularly UV-B (wavelengths 280–315 nm). While moderate UV-B levels do not affect pure photobiont responses, they can substantially reduce growth driven by both symbionts together. It seems likely that the mycobiont is more sensitive to UV-B-than the photobiont, and melanins in lichens protect the fungus from the harmful effects of UV radiation. While not needed for UV protection, melanins have been shown to increase the tolerance of photobionts to photoinhibition by PAR [13]. Melanin-producing lichens include species that grow in shaded habitats e.g., *Lobaria* spp., but also lichens from more exposed sites such as *Cetraria* [14]. Typically, melanised individuals of a particular species can be collected from more exposed microhabitats that are just a short distance (tens of centimetres) from slightly more shaded pale individuals.

The relative importance of the various tolerance mechanisms for photoprotection in different species of lichens in field situations is unknown. In particular, it is difficult to separate mechanisms based on the synthesis of light screening pigments from more biochemical mechanisms of photoprotection. For lichens that become melanised when they grow in sunny locations it appears intuitive that melanin synthesis would be the most important defense mechanism. As noted above, melanised thalli are more tolerant to photoinhibition than pale thalli [13,15]. Unfortunately, the problem with making such simple comparisons between pale and brown thalli is that melanised thalli have a history of exposure to higher light levels than pale thalli. As a result, the photobionts of melanised thalli may have developed other mechanisms that increased tolerance to photoinhibition. Recently, we compared the importance of melanisation in tolerance to high light with the importance of other tolerance mechanisms by dissecting away the lower cortices and medullas in a range of species [16]. This enabled us to photo-inhibit photobionts with light from below i.e., without the presence of a melanised upper cortex. In all species, compared with pale thalli, photobionts in melanised thalli were more tolerant to photoinhibition when exposed from above. However, when photoinhibited from below, the tolerances of photobionts from melanised thalli of *Crocadia aurata* and *Lobaria pulmonaria* were rather similar to those of photobionts from pale thalli. In these species, melanin synthesis does indeed appear to be the main tolerance mechanism. However, in *Cetraria islandica* the tolerance of photobionts in melanised thalli photoinhibited when exposed from below was still significantly higher than that of pale thalli. This suggests that for *C. islandica* protection from high light appears to derive from a mixture of both cortical pigments and biochemical mechanisms.

Apart from the dissection study discussed above [16], there have been few attempts to investigate the importance and nature of other mechanisms of tolerance to high light in pale and melanised thalli of the same species of lichens. When growth is measured in differently melanised thalli of *L. pulmonaria*, results suggested that the mycobiont adjusted to the light received by the photobiont beneath the screening upper cortex to rather uniform levels, for example across a gradient in tree canopy openness [17]. The implication would be that photosynthetic and photoprotective parameters, for example NPQ, should not differ between pale and melanic thalli. Recently, we compared the induction and relaxation of NPQ in melanised and pale thalli of members of three species of shade lichens, including *L. pulmonaria* [10]. Consistent with the study that involved growth measurements [17], in *L. pulmonaria*, NPQ was rather similar in pale and melanised thalli. However, in *Lobaria virens* and *C. aurata* melanised forms have more, and faster relaxing NPQ than pale forms. Interestingly, in a similar study, we also showed that in five species of non-melanising lichens, shade forms generally display higher NPQ than sun forms [10]. For these species, it seems likely that the higher NPQ in the shade forms may protect photobionts from occasional rapid increases in light that occur during sunflecks. As both pale and melanised material of *L. virens* and *C. aurata* grow in relatively shaded habitats, the higher NPQ in the melanised thalli may be induced simply to provide additional protection over and above melanins. The implication is that, unlike *L. pulmonaria*, melanins in *L. virens* and *C. aurata* may not adequately reduce light levels at the photobiont layer. At the moment, the relative importance of NPQ in the photoprotection of melanising species that grow in more open habitats is unknown.

An inherent problem with comparing chlorophyll fluorescence parameters in pale and melanised thalli is that in melanised thalli the photobiont layer is being analyzed through a pigmented upper cortex. As discussed above, our preliminary data suggest that the characteristics of the induction and relaxation of NPQ can differ in melanised and pale thalli from the same species [10]. The widely accepted method of estimating NPQ is based on measuring ratios of fluorescence signals, not absolute values [18]. Therefore, the partial attenuation of the measuring beam by a melanised cortex should in theory not have a great affect on the validity of measurements at the intact thallus level. However, attenuation of the actinic light used for NPQ determination may increase F_M’_ (maximal fluorescence yield of the light-adapted state) and therefore decrease NPQ. In other words, at the level of the photobionts, values of NPQ may be measured at a slightly lower light level. Therefore, our first hypothesis was that the presence of a melanised cortex should not prevent the use of conventional chlorophyll fluorimetry to compare NPQ in pale and melanised thalli. Our aim was to compare the induction of NPQ in melanised and pale thalli of *L. pulmonaria* and *C. aurata* using the dissection technique [16]. Essentially, the lower cortices and medullas were dissected away in pale and melanised thalli, and the induction and relaxation of NPQ measured as a function of time following illumination and subsequent darkness.

Our second aim was to assess the role of NPQ in pale and melanised thalli of the lichens *Peltigera aphthosa* and *C. islandica*. Unlike the species in our earlier study [10], these two lichens grow in rather open microhabitats, i.e., habitats characterized by slow rather than rapid fluctuations in light. Results presented here show that chlorophyll fluorescence can be validly used to compare NPQ in pale and melanised thalli without the need for dissection experiments. However, the dissection technique revealed that there may be subtle differences in the characteristics of the photobionts at the top and bottom of the photobiont layer. When results from the four species tested here are considered together, it appears that no simple relationship exists between melanisation and NPQ.

## 2. Results

Figure 1 illustrates a typical screen shot from the Walz Imaging PAM for lichen disks illuminated from above (A) and from below with the lower cortex and medulla removed (B). Figure 2 compares the induction and relaxation of NPQ and the induction of rETR in pale and melanised collections of *L. pulmonaria* and *C. aurata* illuminated with light at 100 μmol m^−2^ s^−1^ from above and from below with the lower cortex and medulla removed. Results of a repeated measure ANOVA for these data are presented in Table 1. NPQ was induced more rapidly in melanised than pale collections of both species. In *L. pulmonaria* illuminated from above, pale thalli displayed slightly higher NPQ than melanised thalli after 11 min, while in pale and melanised material illuminated from below, NPQ values were similar. The effect of melanisation on NPQ was significant (Table 1). While melanised thalli display a similar induction and relaxation of NPQ whether illuminated from above or below, in pale thalli NPQ was lower when illumination was from below, although overall the effect of position (above or below) was not significant (Table 1). In *C. aurata* NPQ was always higher in melanised than pale thalli, and always higher when thalli were illuminated from above. The effect of melanisation on NPQ was highly significant, and the effect of position significant at *p* < 0.05 (Table 1). For all thalli of both species, the induction of rETR was rapid, and rETR was slightly higher in melanised thalli, although only the effect of melanisation was significant for *L. pulmonaria*, and only the effect of position significant for *C. aurata* (Table 1). Rapid light curves for both species suggested that rETR was slightly higher in melanised forms, and slightly higher when thalli were illuminated from above (Figure 3). Values of rETR obtained when light intensities above 250 μmol m^−2^ s^−1^ were used were anomalous (appearing to rise rapidly) and are not presented here.

Figure 4 compares the induction and relaxation of NPQ and the induction of rETR in pale and melanised collections of *P. aphthosa* and *C. islandica* illuminated with light at 100 μmol m^−2^ s^−1^. For both species, NPQ was higher in melanised than pale thalli. Relaxation of NPQ was fast in *P. aphthosa* but relatively slow in *C. islandica*. Induction of rETR was relatively slow in *P. aphthosa* but rapid in *C. islandica*. In both species, rETR after 11 min was higher in melanised than pale forms. The tendency for melanised thalli to have higher rETR was confirmed by rapid light curves (Figure 5). Melanised thalli of the cyanobacterial lichen *Peltigera malacea*, collected from adjacent to the *C. islandica* tested here, also displayed higher values of rETR. In both *C. islandica* and *P. malacea*, compared with pale forms, melanised forms were apparently more resistant to photoinhibition induced during the construction of the rapid light curves.

## 3. Discussion

Melanisation and NPQ are two mechanisms of photoprotection used by lichens to tolerate long term light stress, although the balance between these two mechanisms in field situations remains unclear. Here we used a dissection technique to show that despite potential problems with measuring lichen photobionts through a variously melanised upper cortex, NPQ can be satisfactorily assessed with a standard fluorimeter. Interestingly, the dissection method revealed that photobionts from the top and bottom of the photobiont layer can display characteristic “sun” and “shade” features, respectively. Taking together the data from all four species studied here, it is clear that no simple relationship exists between melanisation and NPQ.

### 3.1. NPQ Can Be Measured through a Melanised Cortex

It seems likely that lichens protect themselves against the harmful effects of high radiation using a combination of cortical light screening pigments and other, biochemical mechanisms [4]. Recently, we tested the relative importance of melanisation by dissecting away the lower cortex and medulla, and then photo-inhibiting photobionts by light exposure from both above and below [16]. Results showed that for some species, tolerance to photoinhibition was still higher when photobionts were exposed to light in the absence of a melanised upper cortex. This indicates that melanisation is not the only tolerance mechanism present in these species. However, further study of these additional mechanisms of tolerance, e.g., measurement of NPQ could be hindered by the presence of a melanised upper cortex, even though in theory the equation used to calculate NPQ is based on ratios rather than absolute estimates of fluorescence parameters [18]. Unfortunately, our dissection technique is not applicable to all species [16], and furthermore requires the use of an imaging PAM. The first aim of the present study was to test whether valid comparisons of NPQ can be made in collections with and without a melanised upper cortex without resorting to the dissection technique. For example, recently we measured the induction of NPQ as a function of time in pale and melanised collections of two of the species tested here [10]. In *L. pulmonaria* NPQ induced more rapidly in melanised than pale forms but was similar after 11 min in the light. In contrast, in *C. aurata* NPQ induced more rapidly and reached much higher values in melanised than pale forms. The implication is that NPQ may be an important additional tolerance mechanism in melanised *C. aurata* but less so in *L. pulmonaria*. In the collections of *L. pulmonaria* used here, when the photobionts were exposed in the same way as our previous study [10] (i.e., from above, through the upper cortex) the pattern of NPQ induction was similar, although in the present study NPQ after 11 min was slightly higher in pale material (Figure 2). Melanised *L. pulmonaria* exposed from below behaved similarly to material exposed from above, while pale material exposed from below displayed generally lower values of NPQ than material exposed from above. For *C. aurata*, NPQ in lichens exposed from below is generally lower than when exposed from above, but melanised material always displays much higher values of NPQ than material exposed from above. There are likely explanations for the differences in the values of NPQ observed when thalli are exposed from above or below (see next section). Nevertheless, the conclusions on the relative importance of NPQ are similar, whether lichens are exposed from above, or from below, without the presence of a melanised cortex. In other words, results suggest that valuable information about the potential role of NPQ in the tolerance of photobionts to high light can be obtained simply by taking measurements from above using intact thalli with a standard PAM.

### 3.2. Photobionts from the Top of the Photobiont Layer Display Relative Sun Characteristics

Results suggest that differences may exist between the characteristics of photobionts at the tops and bottoms of the photobiont layers in *L. pulmonaria* and *C. aurata* (Figure 2). It can be difficult to estimate how deeply a fluorimeter samples within a leaf or thallus (see [19] for discussion of this point). However, it is probable that the photobionts sampled when the thallus is measured from above differ from those sampled when measurements are made from below, with the lower cortex and medulla removed. In *Cladonia arbuscula* little light penetrates more than c. 30 µm into a thallus, and if this is generally true for other lichens the implication is that the lower parts of the photobiont layer receive less light than those on the top [20]. It could be predicted therefore that the upper parts of the photobiont layer may show more “sun” characteristics while the lower part more “shade” characteristics. Specifically, photobionts at the top of the photobiont layer may display more efficient photosynthesis, but also higher quenching to protect themselves from high light. In the present study, light response curves and rates of rETR after 11 min at 100 µmol m^−2^ s^−1^ were quite similar for pale and melanised thalli, whether measured from above or below (Figure 2). Nevertheless, as predicted, in both species maximum rates of rETR were slightly lower when measurements were made for the lower surface of both pale and melanised thalli (Figure 3). In contrast to rETR, larger differences in NPQ were usually found between the top and bottom of the photobiont layer. Only in melanised *L. pulmonaria* did the induction of NPQ display similar kinetics whether measured from the top or bottom of the thalli. In all other cases, lower values of NPQ were recorded for the lower side. The simplest explanation for this is that shading means that less photoprotection is needed for photobionts in the lower parts of the photobiont layer. There have been very few studies on gradients in photosynthetic acclimation within one lichen thallus. In one study, a microscope imaging PAM was used to study gradients of chlorophyll fluorescence parameters in the desert crust squamulose lichens *Placidium* and *Peltula* and a crustose *Collema* sp. [21]. Gradients in the effective quantum yield of PSII (ΦPSII) were more or less as would have been predicted from the higher plant literature, with the top of the photobiont layers displaying higher efficiency of PSII; however, changes in quenching were not always as predicted. In *Placidium*, qN (the proportion of energy absorbed dissipated as heat) decreased with depth in the thallus, but in *Petula* qN increased, while in *Collema* there was no change. Possibly in squamulose soil crust lichens, reflectance from the substratum may mean that some species receive higher light from below than would be expected. More information is available for higher plant leaves where gradients of light can be quite steep, and are modulated by the presence of pigments, surface waxes and trichomes [19,22]. For example, the chloroplasts isolated from paradermal sections prepared from the leaves of *Camellia japonica* and *Spinacia oleracea* were compared [23,24,25]. A clear gradient in the chloroplast properties from sun to shade exists within these leaves. Components such as rubisco and cytochrome f expressed on a chlorophyll basis were much greater in the sun-type chloroplasts than in shade type. Similar trends in efficiency have been observed in *Quercus* leaves, and further, NPQ is lower towards the lower surfaces of the leaves [26]. Therefore, the trends of photosynthetic efficiency and NPQ reported in higher plants resemble those found here. It is unclear why values for NPQ are rather similar for the tops and bottoms of melanised *L. pulmonaria* thalli, although possibly in melanised samples of this species the algal layer is thinner. However, results presented here suggest that photobionts from the bottom of the photobiont layer tend to have slightly lower rates of photosynthesis and less quenching than those at the top, i.e., display mild shade characteristics.

### 3.3. Melanised Thalli Display Different NPQ Responses Compared with Pale Thalli

Melanised thalli of the four species tested here, compared with pale thalli, can display values of NPQ that are higher (*C. aurata*), similar (*L. pulmonaria*) or lower (*P. aphthosa* and *C. islandica*) (Figure 2 and Figure 4). As discussed in the Introduction, growth measurements of *L. pulmonaria* suggest that the mycobiont appears to adjust the melanins present in the upper cortex so that the photobiont receives uniform light levels, for example across a gradient in tree canopy openness [17]. Consistent with this suggestion, in the present study NPQ and parameters such as rETR were rather similar in pale and melanised thalli of *L. pulmonaria*. However, in the other species tested, differences were found in NPQ when comparing pale and melanised thalli. For melanised thalli of *C. aurata* melanins seem to provide insufficient photoprotection for the extra light they experience in their microhabitat, and as result, they possess additional NPQ (Figure 4). *P. aphthosa* and *C. islandica* were collected from more open habitats than *C. aurata* and *L. pulmonaria*. Interestingly, although our earlier results [9,10] would suggest that in general NPQ relaxes faster in sun than shade lichens, *P. aphthosa* was an exception in that, for no obvious reason, NPQ relaxes quickly on transition to darkness. Melanised thalli from *P. aphthosa* and *C. islandica* display lower maximum values of NPQ than pale, suggesting that their photobionts may experience less light stress than those from pale thalli. The most straightforward explanation is that melanins may have been synthesized following exposure to particularly bright light at one time of the year, and at the time of collection the resulting melanisation was excessive. Once synthesized, melanins cannot be broken down. Higher estimates of rETR in melanised thalli of these species, and in the cyanobacterial lichen *P. malacea* may have been obtained because the PAR used to estimate rETR was higher than the actual PAR reaching the photobiont (Figure 5). However, it is also possible that the higher rETR in melanised forms indicates that they are showing some sun characteristics, and actually use alternative mechanisms of photoprotection in addition to NPQ. Similarly varied results have been reported from other photosynthetic organisms that may produce pale or pigmented leaves. For example, much higher NPQ occurs in dark red collections of the liverworts *Jamesoniella colorata* and *Isotachis lyallii* collected from exposed sites compared with pale green individuals from more shaded sites [27]. Conversely, pigmented leaves in *Ophiopogon*, *Vitus* and *Prunus* all display lower values of NPQ than pale forms [28,29,30]. For these species, the implication is that strong pigments allow leaves to maintain a smaller content of xanthophyll cycle components and depend less on xanthophyll cycle energy dissipation. However, if excessive, pigment-based photoprotection could be predicted to reduce photosynthesis under conditions of non-inhibitory light levels (e.g., [31]). Interestingly, a more wide-ranging study found rather similar values of NPQ in both young red leaves and paler mature forms [32]. In conclusion, as for results from other pigmented or pale photosynthetic organisms, it seems impossible to generalize as to whether in lichens the photobionts of melanised thalli show higher or lower NPQ than pale forms. Presumably, depending on the light history of the microhabitat and presence of other tolerance mechanisms, all combinations are possible.

## 4. Materials and Methods

### 4.1. Lichen Material

Three cephalolichens were used in this study. *Crocodia aurata* (Ach.) Link. was collected from *Leucosidea sericea* trees growing in Afromontane vegetation at Fort Nottingham, South Africa. *Lobaria pulmonaria* (L.) Hoffm. was collected from the bark of oak trees from nemoral boreal vegetation at Langangen, Norway. *Peltigera aphthosa* (L.) Willd. was collected from an exposed locality in nemoral boreal vegetation on the outskirts of Ås, Norway. The chlorolichen *Cetraria islandica* (L.) Ach. and the cyanobacterial lichen *Peltigera malacea* were collected from an exposed site in boreal vegetation on the outskirts of Syktyvkar, Russia. For each site, collections of pale (shaded) and melanic (more exposed) thalli from the same population (a few cm to up to 5 m apart) were made at the same time. The lichen material was dried at room temperature between sheets of filter paper in the laboratory overnight and then stored refrigerated for a maximum of two weeks. One day before each experiment, discs 1 cm in diameter were cut. For *C. aurata* and *L. pulmonaria* under a dissecting microscope a small section (c. 2–4 mm^−2^) of the lower cortex and the adjacent medulla was scraped away using the tip of a Pasteur pipette until the lower part of the continuous green photobiont layer was fully exposed. All discs were then allowed to hydrate overnight on moist filter paper in Petri dishes under dim lighting (5 µmol m^−2^ s^−1^) at 15 °C in a thermostatically controlled chamber. For all the other species, disks were cut from pale and melanised thalli, and hydrated in the same way. Chlorophyll fluorescence parameters were measured the following day.

### 4.2. Chlorophyll Fluorescence

For *C. aurata* and *L. pulmonaria* chlorophyll fluorescence was measured using the “maxi” red version of the Imaging PAM fluorimeter (Walz, Effeltrich, Germany). At each sampling event, an image of each disc was captured. In the intact discs, fluorescence parameters were integrated over a large area. For the scraped discs, parameters were integrated over the area from which the lower cortex had been removed, typically several mm^2^ in size. Typical images are shown in Figure 1. For all the other species chlorophyll fluorescence was measured using a PAM 2500 fluorometer (Walz, Effeltrich, Germany) using the red LED throughout. As quenching in the cyanobacterial *P. malacea* cannot be measured with a conventional PAM, only rapid light curves were constructed.

After a dark adaptation period of at least 10 min, the maximal efficiency of photosystem II (PSII; F_V_/F_M_) was measured using a 0.8 s pulse of actinic light at 8200 µmol m^−2^ s^−1^, where F_M_ = maximum fluorescence and F_V_ = variable fluorescence or (F_M_ − F_O_), with F_O_ = minimal fluorescence yield of the dark-adapted state. Thalli with anomalous values of F_V_/F_M_ were discarded. Rapid light response curves of relative electron transport rates (rETR) were measured by increasing the actinic light in small steps for 10 to 20 s at each light level from 0 to 250 μmol photons m^−2^ s^−1^ for *L. pumonaria* and *C. islandica* and from 0 to 578 μmol photons m^−2^ s^−1^ for *P. aphthosa*, *C. islandica* and *P. malacea*, with saturating flashes at the end of exposure to each light level. The rETR was calculated as:
rETR = 0.5 × ΦPSII × PAR
where PAR = photosynthetically active radiation and ΦPSII is the effective quantum yield of PSII photochemistry calculated as (F_M’_ − F_t_)/F_M_ (where F_M’_ = maximal fluorescence yield of the light-adapted state and F_t_ = stable fluorescence signal in the light). Equal distribution of excitation between PSII and PSI is assumed by using 0.5.

To determine the induction of rETR, and the induction and relaxation of NPQ, thalli were dark-adapted for 10 min and F_V_/F_M_ measured; thalli with anomalous values were discarded. An actinic light of 100 μmol photons m^−2^ s^−1^ was then turned on, and saturating flashes applied at increasing intervals for 11 min. The actinic light was then turned off and relaxation measured for 10 min, with saturating flashes given at increasing intervals. NPQ was calculated using the formula [18]:NPQ = (F_M_ − F_M’_)/F_M’_

In initial experiments we tested the induction of NPQ using a variety of light intensities, but in a laboratory setting values much above 100 μmol photons m^−2^ s^−1^ tended to cause photoinhibition in some species.

### 4.3. Statistical Analysis

The statistics package “Statistica” (Basic Academic Bundle V14, TIBCO Software Inc., Palo Alto, CA, USA) was used to carry out generalized mixed linear models (repeated measure) for NPQ and rETR in discs of *L. pulmonaria* and *C. aurata* exposed to light at 100 μmol m^−2^ s^−1^ for 11 min followed by darkness for 10 min.

## 5. Conclusions

From the literature on free-living algae and higher plants, it is becoming increasingly clear that photosynthetic organisms use a great diversity of mechanisms to protect themselves against the potentially harmful effects of excess light [11]. A uniquely lichenological mechanism is the production of light-screening melanins by the fungi that form the upper cortex [13]. In addition, what appears to be classical NPQ has been widely reported in lichens [4]. The present study suggests that no simple correlation exists between melanisation and NPQ. In all species tested here, rETR was higher in melanised than pale forms. This is probably at least in part because the photobionts in melanised thalli may receive less light due to screening. Therefore, rETR will be overestimated because it is based on a higher PAR than the photobionts actually receive. In some species, NPQ is similar in pale and melanised forms, suggesting that mycobionts can adjust the melanins present in the upper cortex so that the photobiont receives uniform illumination. In other species, melanisation appears to offer insufficient photoprotection, and the photobionts display additional NPQ. In still other species, NPQ is actually less in melanised than pale forms, suggesting that either melanisation may have been excessive earlier in the growing season, or alternatively that these species use other mechanisms to protect themselves from the effects of high light. In these species there is a clear need to investigate other potential mechanisms of tolerance to high light stress.

## Figures and Tables

**Figure 1 plants-11-02726-f001:**
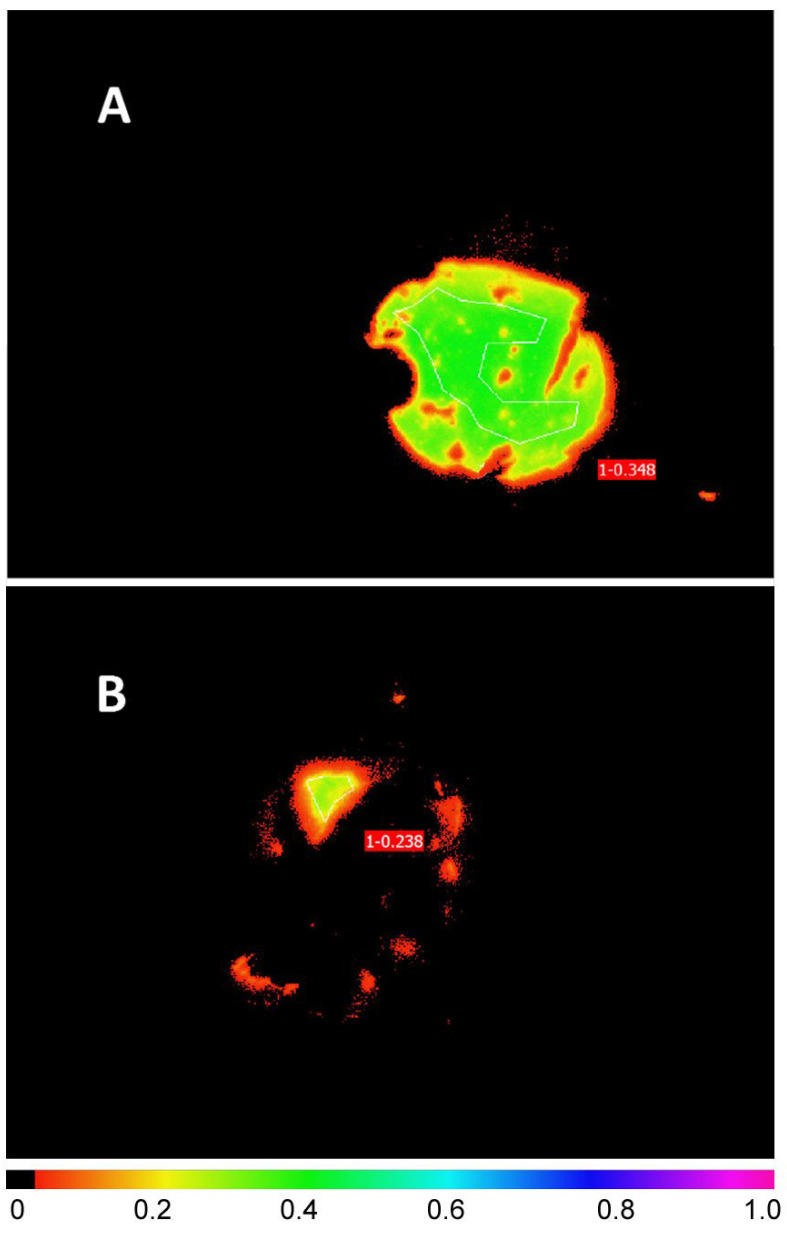
Typical screenshot from the Walz Imaging PAM fluorimeter. Here a melanised disk of *Crocodia aurata* was measured from above (**A**) and melanised disk of *C. aurata* with the lower cortex and medulla partially removed was measured from below (**B**). The disks were 1 cm in diameter. The parameter illustrated here with false colour is F_M’_ (=maximal fluorescence yield of the light-adapted state). The white lines show the areas of interest used for calculation of the F_M’_ values shown.

**Figure 2 plants-11-02726-f002:**
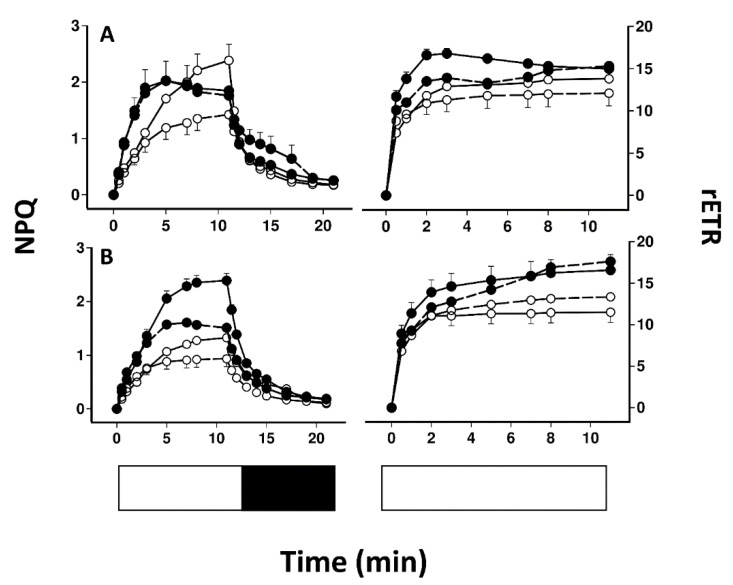
The effect of melanisation on the induction and relaxation of NPQ and the induction of rETR following exposure to light at 100 μmol m^−2^ s^−1^ in *Lobaria pulmonaria* (**A**) and *Crocodia aurata* (**B**). Open symbols indicate pale thalli and solid symbols melanised thalli. Solid lines represent intact material exposed to high light from above, while dashed lines represent material with part of the lower cortex and medulla removed and illuminated from below. Error bars show the mean 1 ± SE (*n* = 10) when larger than symbol size. The white section in the row at the base of each graph indicates the time periods when samples were exposed to light, and the dark when samples were exposed to dark. Statistical analyses of these graphs are present in Table 1.

**Figure 3 plants-11-02726-f003:**
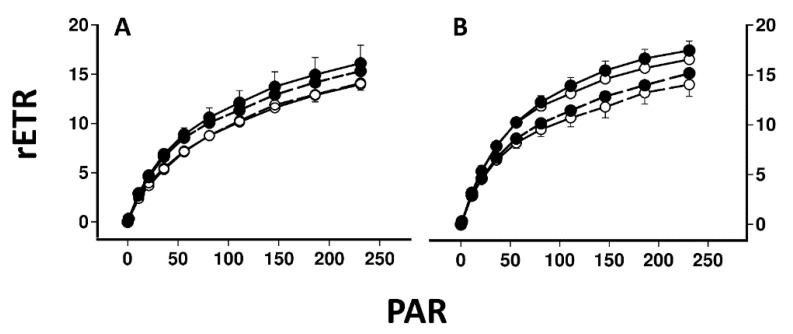
Rapid light curves (rETR as a function of light level) in *Lobaria pulmonaria* (**A**) and *Crocodia aurata* (**B**). Symbols and lines as for Figure 2. Error bars show the mean 1 ± SE (*n* = 10) when larger than symbol size.

**Figure 4 plants-11-02726-f004:**
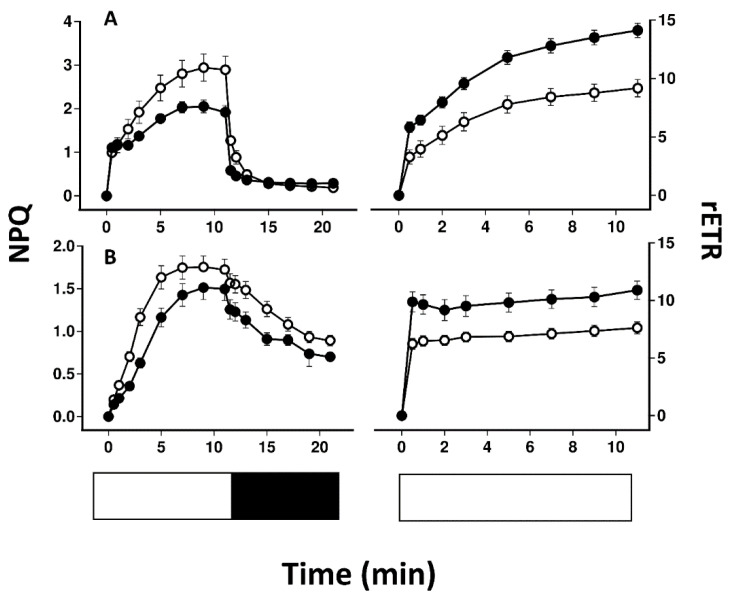
The effect of melanisation on the induction and relaxation of NPQ and the induction of rETR following exposure to light at 100 μmol m^−2^ s^−1^ in *Peltigera aphthosa* (**A**) and *Cetraria islandica* (**B**). Symbols and lines as for Figure 2. Error bars show the mean 1 ± SE (*n* = 10) when larger than symbol size. The white section in the row at the base of each graph indicates the time periods when samples were exposed to light, and the dark when samples were exposed to dark.

**Figure 5 plants-11-02726-f005:**
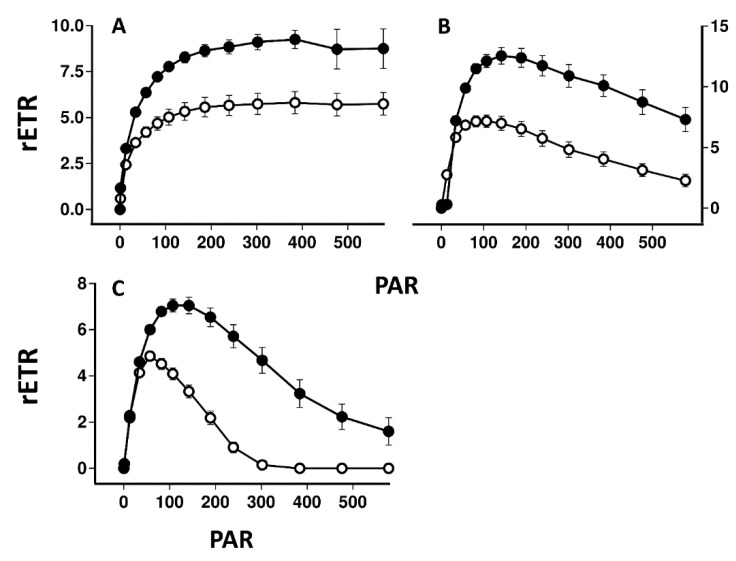
Rapid light curves (rETR as a function of light level) in *Peltigera aphthosa* (**A**), *Cetraria islandica* (**B**) and *Peltigera malacea* (**C**). Symbols and lines as for Figure 2. Error bars show the mean 1 ± SE (*n* = 10) when larger than symbol size.

**Table 1 plants-11-02726-t001:** Generalized mixed linear models (repeated measured) for NPQ and rETR in discs of *Lobaria pulmonaria* and *Crocodia aurata* exposed to light at 100 μmol m^−2^ s^−1^ for 11 min followed by darkness for 10 min. Fixed factors: melanisation (M, pale versus melanic); position (P, exposed from above through an intact upper cortex or from below with the lower cortex and medulla removed); time (T) during exposure to light and then darkness. Interactions between the factors are indicated with as e.g., M x P, meaning the interaction between melanisation and position. There were ten replicates for each treatment combination. *** *p* < 0.001; ** *p* < 0.01; * *p* < 0.05.

Effect	Degrees of Freedom for NPQ	*Lobaria* NPQ	*Crocodia* NPQ	Degrees of Freedom for ETR	*Lobaria* rETR	*Crocodia* rETR
Melanisation (M)	1	*	***	1	***	0.202
Position (P)	1	0.206	*	1	0.057	*
M x P	1	0.081	0.337	1	0.163	0.312
Time (T)	15	***	***	7	***	***
T x C	15	***	***	7	***	**
T x P	15	***	***	7	***	***
T x C x P	15	***	0.059	7	0.411	***

## Data Availability

Not applicable.

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
