# Peer review of "Melanisation in Boreal Lichens Is Accompanied by Variable Changes in Non-Photochemical Quenching"

_plants, 2022, doi:10.3390/plants11202726_

Round 1

Reviewer 1 Report

In this work, Truelove Ndhlovu et al analyze the non-photochemical quenchinig (NPQ) parameter in two lichen species and the correlation of the values of this parameter with the presence or absence of pigmentation (melanine) in the talli of both species.

They show that the pigmentation does not affect to the determination of NPQ and that there is not correlation between the level of pigmentation and the values of NPQ, which can be higher, similar or lower in pigmented talli compared with pale talli.

The manuscript is clearly written, the experiments well designed and the discussion is supported by the data obtained.

Some comments:

-       I do not understand Table 1. It should be explained more clearly. For instance, what freedom for NPQ means?, What is rETR?, What is T*C or T*C*P? among others

-       The phrase in lines 147-151 is rather confusing. It says that the NPQ  is different when talli is illuminated from above or below, but in the next sentence they say that the effect of position (above or below) was not significant

-       In Figure 4 an (A) seems to be lost after Peltigera aphthosa.

Reviewer 2 Report

I’ve read the manuscript “Melanisation in boreal lichens is accompanied by variable 1 changes in non-photochemical quenching” and my comments are summarised in the following lines.

In general the introduction should focus more on the role of melanin in photoprotection, its biosynthesis and presence in lichens. In particular the its role in UVB protection should be described here and discussed later. In this regard the paper  by Solhaug et al New Phytologist (2003) 158: 91–100 should be cited.

The purpose and hypotheses of the study are not clearly stated in the introduction, and it is easy to get lost in the description of the results.

Nothing is said about dark relaxation kinetics and photoinhibitory quenching that have been measurent in the NPQ curves.

It is difficult to compare fluorescence responses in melanised and non melanised thalli. Melanine absorbs a certain amount of light, so light reaching the photobiont layer differs among samples. This effect is probably small for the case of red light, but this has to be discussed. This point should be discussed.

Specific comments:

-line 13, unclear sentence: lichen photobionts do not synthesise light screening compounds.

-line 15, more specifically the problem is that light reaching the photobiont layer varies among individuals making difficult the comparison of fluorescence parameters.

-line 21, rates of photosynthesis have not been measured, just apparent electron transport. There are other electron sinks apart from CO2 assimilation.

-line 42, there are several types of quenching mechanisms and NPQ has more than two main components (qE, qZ, qI, qT,…).

-line 56, additionally lichens show a type of desiccation induced NPQ

-Line 82, is the main role of melanin protection against visible light or against UV? How is its absorption spectrum?

-Line 124, it could affect if saturation pulses do not reach photobiont layer with enough intensity as to close reaction centres and to obtain a genuine Fm (or Fm’) value.

-Results: Scientific names have to be written in italics throughout the whole section.

-Figure1 is not necessary, it is more informative to show an image illustrationg  the dissection technique

-Legend of Fig 2, line 183, is this sentence correct? do error bars represent the mean??

-line 246-246. Unclear sentence.

-line 354, conditions (temperature, RH,…) during desiccation of thalli.

- It’s difficult to compare NPQ in melanised and pale forms, because both differ in  light intensity reaching photobionts

-would it be possible to remove the upper cortex?

-What is the light intensity of the saturating pulses?

Round 2

Reviewer 2 Report

I still consider that Figure 1 is unnecessary. If the authors want to show examples of chlorophyll fluorescence images, the screenshots can be combined with regular pictures of the same thalli, otherwise readers won't understand what the figure shows.

Author Response

Dear Editor, one reviewer still questions the inclusion of Figure 1, which illustrates chlorophyll fluorescence images of the lichen disks. I have chatted with the other authors, and we remain convinced that the inclusion of these images helps the reader to understand exact what we did in our "dissection technique". This technique was to remove the lower cortex and medulla so that the photobiont layer could be exposed from the under side of the thallus, and the figure illustrates exactly this. However, we do not wish this to be a "sticking point" that delays acceptance, so if requested by the editor we will happily delete the figure. Inclusion of regular photographs of the thallus we feel would not help to understand the procedure.